# Velocity Increment on Incidence Angle near the Leading Edge of the Compressor Cascade

**Xiaobin Xu** [1,2], **Baojie Liu** [2,3], **Xianjun Yu** [2,3] and **Guangfeng An** [2,3,*]

1   School of Energy and Power Engineering, Beihang University, Beijing 100083, China
2   National Key Laboratory of Science & Technology on Aero-Engine Aero-Thermodynamics, Beihang University, Beijing 100083, China
3   Research Institute of Aero-Engine, Beihang University, Beijing 100083, China
*   Correspondence: guangfeng@buaa.edu.cn

**Abstract:** The geometry of a compressor leading edge has an important effect on the aerodynamic performance at an off-designed incidence angle. The current geometric design methods of the leading edge are usually developed based on the flow characteristics at the designed incidence angle. However, few research focuses on the quantitative rules of the leading edge flow characteristics at the off-designed incidence angle in a compressor cascade. This situation restricts the further optimization and development of the leading edge geometry design method. In this paper, starting from the research of a potential cascade theory, the singularity point, where the surface velocity approaches infinity in the leading edge region, is eliminated by applying the characteristic that the ratio of the velocity increasement on the incidence angle in the plate cascade and the isolated plate flow is finite. Secondly, the equivalent pitch lengths based on $1/\cos(\beta)$ and VI caused by a diffuser deceleration in the cascade passage were employed to correct the effect of the stagger angle. Finally, by introducing the isolated flow around the thick airfoil and considering the influence of the camber line geometry, a model of the variation of the surface velocity near the leading edge under the off-designed incidence angle, named the velocity increment on incidence angle, is derived from any compressor cascade. Hence, the relation between the off-designed incidence angle and the designed incidence angle of the surface velocity in a cascade blade is established, and it depends only on the geometrical parameters. Through a verification using numerical calculations and experimental measurement, the explicit formula for the velocity increment on incidence angle proposed in this paper has high precision near the leading edge.

**Keywords:** compressor cascade; potential theory; off-designed incidence angle; surface velocity





## 1. Introduction

The geometry of a leading edge has an important effect on the aerodynamic performance of the axial flow compressor blade at an off-designed incidence angle, such as that showed by Carter in his experimental study whereby the smaller the blade leading edge size, the smaller the blade loss and the larger the operational incidence range [1]. Unlike the circular leading edge, which is widely used, Walraevens and Cumpsty confirmed that the boundary layer behind the elliptic leading edge was healthier, and they indicated that the difference was due to the laminar separation bubbles caused by the velocity spike around the leading edge [2]. With the deepening of the research, Shi et al. indicated that the entire integral entropy generation in the boundary layer increased sharply when the laminar separation bubble moved upstream to the leading edge [3], and Lambert et al. observed that the size of the laminar separation bubble increased with a larger incidence angle [4]. Given the importance of a leading edge spike, Goodhand et al. quantified the spike intensity [5], and they presented a continuous-curvature leading edge to eliminate a spike at the designed incidence angle. Following this research, Xiang et al. [6] discussed

that the continuity of the first derivative of curvature can restrain the separation of the boundary layer and delay the transition process. Liu et al. [7] and Song et al. [8] also confirmed that a well-designed continuous-curvature leading edge can eliminate spikes at the designed incidence angle and improve the blade performance by using the CST method and B-spline method, respectively.

The current problem of the blade leading edge research is how to generate the best geometry with the largest operational incidence range. The current geometry generation methods are usually applied for the single designed flow condition, then, the off-designed aerodynamic performance is obtained by a numerical method, and finally, the optimal geometry with the targeted composite performance could be selected after several iterations [9]. Many optimization methods have been proposed, such as, a genetic algorithm [10], particle swarm optimization [11], a surrogate model with a response surface [12], deep learning [13] or reinforcement learning [14], et al. However, in those methods, the off-designed aerodynamic performance appears passively, and more resources are consumed in the iterations. A multipoint inverse design method posed by Michael is another alternative road, and it considers the ideal velocity distribution on the blade surface under different working conditions [15,16]. However, their generation of airfoil still relies on iterations.

Unlike the increasingly complex optimization methods, the theoretical innovation could simplify the blade generation process. However, for the blade under off-designed incidence angles, there are no explicit quantitative rules of the flow characteristics for a universal compressor cascade. If the off-designed performance could be actively embodied in the design procedure by utilizing the aerodynamic relation between the off-designed and the designed flow condition, the iterations in the process of leading edge geometry generation can be reduced or even avoided. However, in current studies, the surface velocity under the off-designed incidence angle is usually obtained from CFD results or implicit potential flow solution. The rise of the CFD method has brought about a large number of flow field details, but these flow field details cannot be used to explore the general rules under different flow conditions. Before CFD was applied in engineering, the analytical conformal transformation method based on potential flow theory was widely used as a flow analysis tool in blade profile design [17,18]. However, the potential theories for cascade flow are all implicit equations and have remained stagnant for decades.

To understand the basic mathematical rules of the cascade flow field in more detail, the potential flow theory with explicit equations for cascade should be proposed. Recently, along with the development in applied mathematics, the integral problem with boundary conditions was applied to the cascade flows. In this way, Baddoo and Ayton obtained an explicit solution for the cascade flows with a small stagger angle, small turning angle, thin thickness, and sharp trailing edge [19]. The explicit solution can directly reflect the mathematical rules of various geometric parameters on the flow field in the compressor cascade, but the accuracy is poor near the blade leading edge because of the infinity tendency at the leading edge point. Moreover, the explicit solution should be extended to a real cascade blade with a larger stagger angle and larger thickness.

This paper will study the steady, incompressible, non-viscous and non-rotational potential cascade flow theory, supplemented with numerical and experimental verification, in order to find the aerodynamic relation between the off-designed and the designed flow condition. Finally, the explicit mathematical rules of the surface velocity increasement on the incidence angle for universal compressor cascades would be presented.

## 2. Potential Analysis of Simplified Cascade

For the two-dimensional cascade of general incompressible axial compressor studied in this paper, different values can be selected for pitch length, $t$, inlet metal angle, $\beta_1$, outlet metal angle, $\beta_2$, camber line geometry and thickness distribution when the chord length, $c$, is specified. The cascade geometry and parameter definitions are shown in Figure 1, in which $\alpha_1$ is the inlet flow angle, $\alpha_2$ is the outlet flow angle, $\beta$ is the stagger angle, and the incidence angle is defined as $i = \alpha_1 - \beta$. When the velocity on the blade surface is

represented by a vector, the positive direction flows clockwise. To consider the high degree of freedom in cascade geometry, the study of the general rules of universal cascade should first simplify the two-dimensional cascades properly, and then add the influence of different geometry parameters into the simplified model.

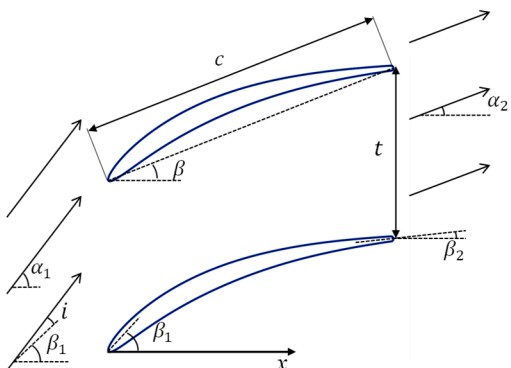

**Figure 1.** Geometry parameters of the two-dimensional planer cascade.

In this paper, the study of plate cascades will be firstly introduced and then extended to elliptic straight cascades with stagger angles. Finally, the influence of actual thickness distribution and camber line geometry will be considered to form a model of the variation of the surface velocity near the leading edge under the off-designed incidence angle, which is derived from any compressor cascade. Furthermore, the flow fields calculated with the conformal transformation method would be treated as the true value, as they have mathematical accuracy in potential flow [20].

### 2.1. Potential Analysis of Plate Cascade

From Baddoo and Ayton's study [19], when the trailing edge always conforms to the Kutta condition, the explicit expression of the influence of the incidence angle on the axial velocity of the blade surface is shown in the following equation:

$$u_i^{\pm}(x) = \mp iU_\infty e^{-\frac{\pi}{t}} \sqrt{\frac{\sinh(\pi(1-x)/t)}{\sinh(\pi(1+x)/t)}} \tag{1}$$

where, the value of $x$ is $-1$ to $1$, hence, the chord length is always 2. Since the influence of the stagger angle and thickness was ignored in the derivation procedure, Equation (1) is essentially the relationship of the surface velocity of the cascade blade on the incidence angle, the pitch length, and the axial position. When the pitch length tends to infinity, Equation (1) is equivalent to the incidence term of the flow around an isolated plate [21]:

$$u_i^{\pm}(x) = \mp iU_\infty \sqrt{\frac{(1-x)}{(1+x)}} \tag{2}$$

Furthermore, the error of Equation (1) in a general cascade is $\mathcal{O}(\varepsilon^2)$, where $\varepsilon$ is approximately equal to the ratio of thickness height to chord length.

In this paper, consider the definition of the velocity increment on incidence angle, VI, as the derivative of the velocity to incidence angle, as described in the equation:

$$VI = \frac{1}{U_\infty}\frac{du}{di} \tag{3}$$

It shows the change in blade surface per unit incidence angle (in radians). Hence, VI of the plate cascade follows the equation:

$$VI_{plate,cascade} = e^{-\frac{\pi}{t}} \sqrt{\frac{\sinh(\pi(1-x)/t)}{\sinh(\pi(1+x)/t)}} \tag{4}$$

and VI of the isolated plate follows the equation:

$$VI_{plate,isolated} = \sqrt{\frac{(1-x)}{(1+x)}} \tag{5}$$

The VI distribution of plate cascade with axial position x under different pitch length is shown in Figure 2. For VI at any pitch length, the value is maximum at the leading edge point and tends to infinity, and then decreases rapidly at the axial position toward the trailing edge. When the pitch length tends to infinity, the value of Equation (4) is completely equivalent to the value of Equation (5).

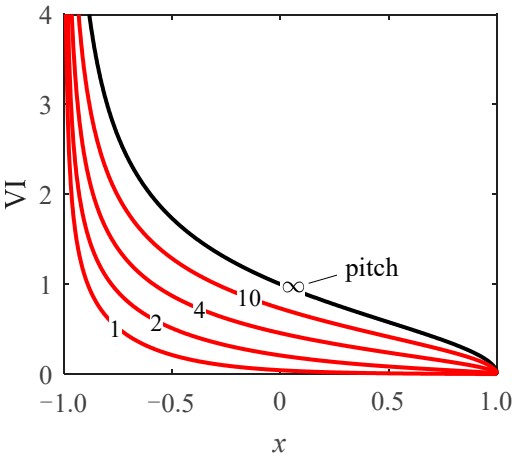

**Figure 2.** Value of VI for plate cascade with different pitch length and axial position.

### 2.2. Potential Analysis of Elliptic Straight Cascade

Evidently, VI at the leading edge point of any thick blade will not take an infinite value, which shows the application limitations of the analytical explicit expressions shown in Equations (4) and (5). However, the ratio of VI in the plate cascade to VI on an isolated plate, $VI_{plate,cascade}/VI_{plate,isolated}$, maintains to be finite, as shown in Figure 3. Hence, the defect whereby the current explicit analytical solution cannot be applied to the leading edge region has been solved.

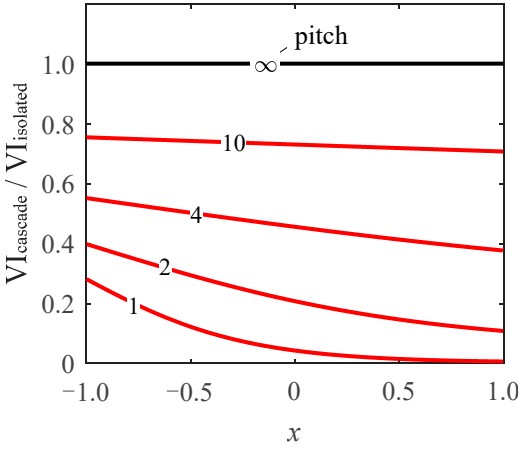

**Figure 3.** Ratio of VI in the plate cascade to VI on an isolated plate.

For the flow around an ellipse with a major axis of 2 on the $x$ axis, the surface velocity under the Kutta condition follows the equation:

$$U(\theta) = \frac{2U_\infty \sin\left(\dfrac{\theta}{2}\right)\cos\left(\dfrac{\theta}{2} - i\right)(\text{AR}+1)}{\sqrt{\text{AR}^2\sin^2(\theta) + \cos^2(\theta)}} \tag{6}$$

in which $\theta$ is the angular parameter (the axial position $x = \cos(\theta)$) and AR is the axial ratio of the ellipse. Hence, VI of the isolated ellipse could be expressed as:

$$\text{VI}_{\text{ellipse,isolated}} = \frac{2\sin\left(\dfrac{\theta}{2}\right)\sin\left(\dfrac{\theta}{2} - i\right)(\text{AR}+1)}{\sqrt{\text{AR}^2\sin^2(\theta) + \cos^2(\theta)}} \tag{7}$$

and VI of the elliptic straight cascades could be established from the flow around the isolated ellipse by utilizing the relationship between plate cascade and flow around the isolated plate, which is expressed as:

$$\text{VI}_{\text{ellipse,cascade}} = \frac{\text{VI}_{\text{plate,cascade}}}{\text{VI}_{\text{plate,isolated}}} \cdot \text{VI}_{\text{ellipse,isolated}} \tag{8}$$

The comparison between the value obtained from Equation (8) for an elliptical cascade with $t = 2$ and the true value is shown in Figure 4. In Figure 4a, the inferred value is almost the same to the true value; it shows the availability of Equation (8). Figure 4b shows the specific difference between the two values with different axial ratios; it indicates that the leading edge region has the largest difference, and the closer the blade surface position is to the trailing edge, the smaller is the difference. At the same time, with the same pitch length, the smaller the axial ratio of the ellipse, the larger the difference. However, the axial ratio does not affect the difference at the leading edge point. The errors between the inferred value and true value are not larger than $\mathcal{O}(\varepsilon^2)$; therefore, the assumption made by Equation (8) is reasonable.

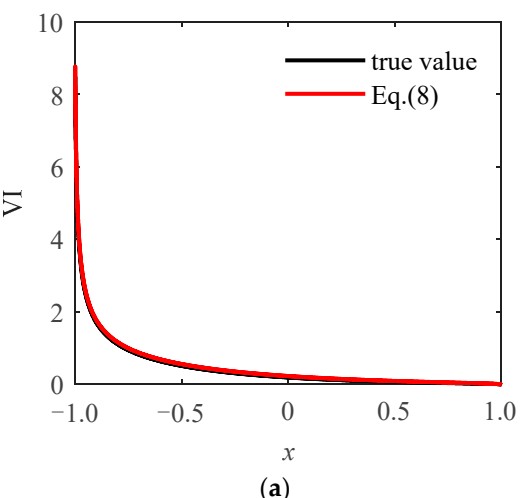

(**a**)

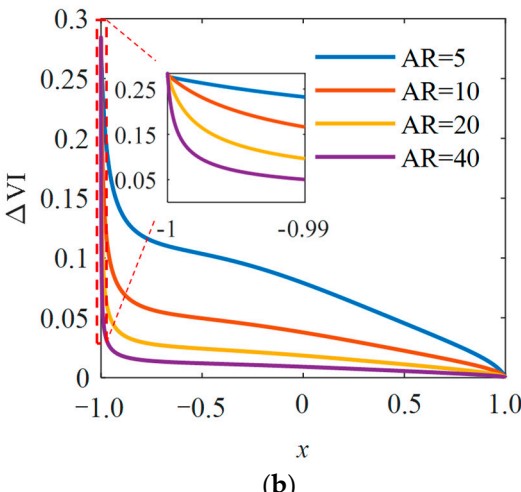

(**b**)

**Figure 4.** True value and the value from Equation (8) of VI in an elliptic straight cascade. (**a**) $t = 2$, AR = 10, (**b**) $t = 2$.

### 2.3. Potential Analysis of Elliptic Straight Cascade with Stagger Angle

For the elliptic cascade with $t = 2$, AR = 10, the effect of the stagger angle is shown in Figure 5, while the inlet flow angles change, following the stagger angles to keep the front stagnation point on the leading edge point. Figure 5a indicates that the larger the stagger

angle, the larger the value of VI near the leading edge region, and the larger the stagger angle is, the larger the difference between VI values of the pressure surface and suction surface in the blade body. Evidently, this is the influence of the change of inlet flow angle under the large stagger angle, which leads to the relative expansion and deceleration of the flow passage. The distribution of VI along the blade surface forms a closed ring, and the suction surface branch is always located in the lower half of the closed ring, that is, VI is always positive near the leading edge of the suction surface; the back half is negative, and the pressure surface is always positive. While the surface velocity is scalar, the distribution of VI is plotted in Figure 5b. In the figure, the upper part is the suction surface side, and the lower part is the pressure surface side. That is, with the increase in the incidence angle, the absolute velocity of the pressure surface always decreases, while the velocity of the suction surface increases in the leading edge region and decreases in the latter part of the blade.

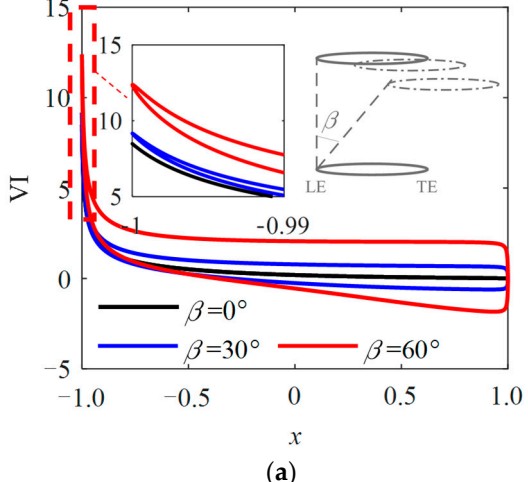

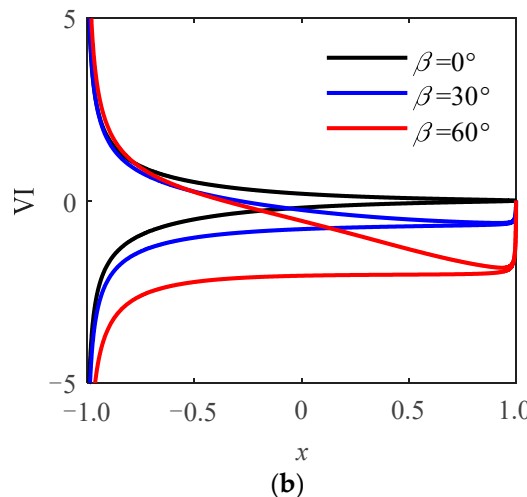

(**a**)  (**b**)

**Figure 5.** VI distribution of an elliptic straight cascade with different stagger angles under *t* = 2, AR = 10. (**a**) VI calculated from vector velocity, (**b**) VI calculated from scalar velocity.

### 2.3.1. Equivalent Pitch Length on Stagger Angle

The method proposed by Baddoo and Ayton to correct the stagger angle is to consider only the change of chord line projection [19], which has a large error in the actual cascade. However, using the equivalent pitch length in:

$$t_{\mathrm{eq}} = \frac{t}{\cos(\beta)} \tag{9}$$

we can correct the influence of the stagger angle near the leading edge, as shown in Figure 6. In that figure, the corrected value of VI on the leading edge point matches the true value well, and the correction error is also smaller than $\mathcal{O}(\varepsilon^2)$. Furthermore, the method proposed by Baddoo and Ayton [19] introduces the opposite correction. Therefore, the correction of Equation (4) changes to:

$$\mathrm{VI}'_{\mathrm{plate,cascade}} = e^{-\frac{\pi}{t_{\mathrm{eq}}}} \sqrt{\frac{\sinh\left(\pi(1-x)/t_{\mathrm{eq}}\right)}{\sinh\left(\pi(1+x)/t_{\mathrm{eq}}\right)}} \tag{10}$$

and the correction of Equation (8) changes to:

$$\mathrm{VI}_{\mathrm{stagger,corr}} = e^{-\frac{\pi}{t_{\mathrm{eq}}}} \sqrt{\frac{\sinh\left(\pi(1-x)/t_{\mathrm{eq}}\right)}{\sinh\left(\pi(1+x)/t_{\mathrm{eq}}\right)}} \cdot \sqrt{\frac{(1+x)}{(1-x)}} \cdot \frac{2\sin\left(\frac{\theta}{2}\right)\sin\left(\frac{\theta}{2}-i\right)(\mathrm{AR}+1)}{\sqrt{\mathrm{AR}^2\sin^2(\theta)+\cos^2(\theta)}} \tag{11}$$

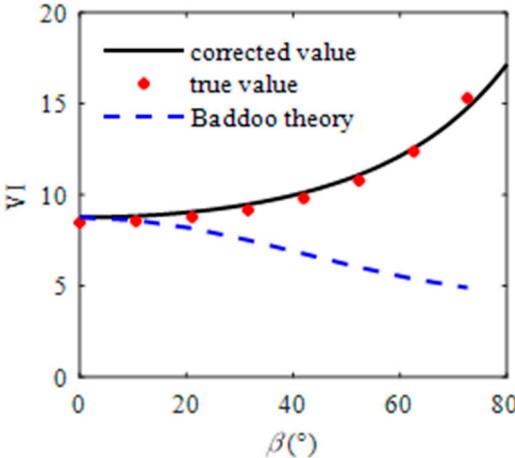

**Figure 6.** Stagger effect on VI at leading edge, $t = 2$, AR = 10 [19].

2.3.2. Diffuser Deceleration on Stagger Angle

The outlet velocity of cascade is only related to the inlet and outlet flow angle, and the three are constrained by the flow conservation equation:

$$U_{\text{out}} = U_{\text{in}} \frac{\cos(\alpha_1)}{\cos(\alpha_2)} \tag{12}$$

under the incompressible assumption. Due to the Kutta condition, the rear stagnation point of the flow is always at the trailing edge point, so the outlet flow angle almost does not change with the inlet flow angle within a certain range. Therefore, when the inlet flow angle changes, the diffuser degree of the flow passage changes. In this case, the derivative of the outlet velocity changing with the inlet angle of attack satisfies:

$$\frac{dU_{\text{out}}}{U_{\text{in}}di} = \text{VI}_{\text{TE}} = -\frac{\sin(\alpha_1)}{\cos(\alpha_2)} \tag{13}$$

namely, the expression of VI at the trailing edge point. Considering the vector representation of blade surface velocity, the $\text{VI}_{\text{TE}}$ of pressure surface and suction surface should be equal in magnitude and opposite in sign.

In the body region from the leading edge to the trailing edge, the average local flow angle is between the inlet and outlet flow angles. As the turning of airflow needs the driver of pressure difference and the velocity increasement represents the velocity difference driven by the pressure difference, the ratio of the flow turning angle to the inlet flow angle in the body region could be described with:

$$T(x) = 1 - \frac{\int_{-1}^{x} \text{VI}_{\beta,\text{corr}}(s)ds}{\int_{-1}^{1} \text{VI}_{\beta,\text{corr}}(s)ds} \tag{14}$$

In this definition, at the leading edge, $T(-1) = 1$, and at the trailing edge, $T(1) = 0$. Therefore, VI caused by a diffuser deceleration, caused by the changing of the inlet flow angle, satisfies:

$$\text{VI}_{\text{diffusion}} = -\frac{[1 - T(x)]\sin(\alpha_1)}{\cos(\alpha_2)} \tag{15}$$

Through the above analysis, the expression of blade surface $\text{VI}_{\beta}$ of an elliptic cascade with a stagger angle is shown as:

$$\text{VI}_{\beta} = \text{VI}_{\text{stagger,corr}} + \text{VI}_{\text{diffusion}} \tag{16}$$

When $t = 2$, AR = 10, $\beta = 30°$, the calculated value of VI in the elliptic cascade has a good matching with the true value, as shown in Figure 7a. Figure 7b shows the distribution of the estimated error of Equation (16) under different stagger angles. It can be seen that the error of Equation (16) gradually increases with the increase in the stagger angle, but the error is always at least one order of magnitude smaller than the value of VI when the stagger angle is not too large. Therefore, Equation (16) can be used as the estimation of VI of the cascades with a stagger angle.

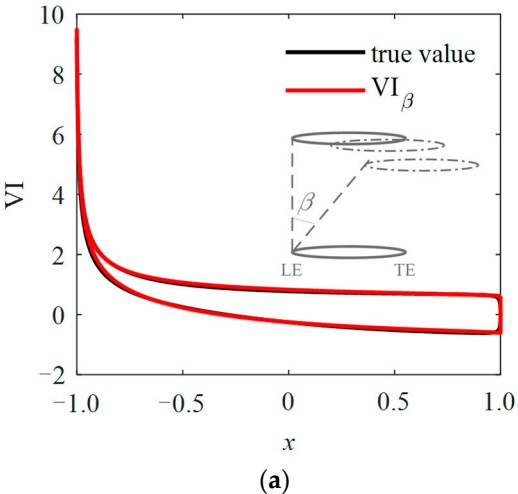

(**a**)

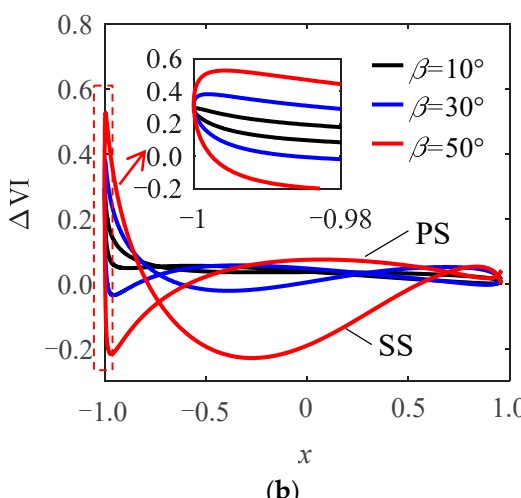

(**b**)

**Figure 7.** Estimated VI of the elliptic straight cascades with stagger angle based on Equation (16). (**a**) $t = 2$, AR = 10, $\beta = 30°$, (**b**) $t = 2$, AR = 10.

It can be seen that diffuser deceleration has a greater impact on the blade body, while the impact on the leading edge is not significant. Therefore, Equation (16) proposed in this paper has been able to achieve a high-precision estimation of the leading edge region.

## 3. Extension in Actual Compressor Cascade

### 3.1. Coordinate System on the Blade Surface

In the above studies, the abscissa of VI is the axial position, while the leading and trailing edge points of the ellipse are located on the $(-1, 0)$ and $(1, 0)$, respectively. When the coordinates are out of the range $[-1, 1]$, Equation (16) would be unsolvable. In the actual blade with a significant curvature in the camber line, the normalization position value should be in the section, $[0, 1]$, strictly, to adapt the x-axis value of $[-1, 1]$ in Equation (16). However, the x-axis value or the normalized chordwise value of the points near the leading edge could be smaller than 0, and the normalized arc length coordinate value then has a discontinuity of derivative near the leading edge. Therefore, the camber projection normalization method is used to define the normalized position of any blade position. As shown in Figure 8, the arc length on the blade surface between any point and the leading edge point is $s$; the arc length on the camber line between the projection point and the leading edge point is $s_\mathrm{p}$; the arc length on the blade surface between the leading edge point and the trailing edge point is $S_0$; and the arc length on the camber line between leading edge point and the trailing edge point is $s_\mathrm{camber}$. Hence, the normalized camber projection coordinate, $s_c$, is described as $s_c = s_\mathrm{p}/s_\mathrm{camber}$, and the normalized surface arc length coordinate is written as $s/S_0$. As the normalized surface arc length coordinate has a higher proportion near the leading edge, it is used in the experiment to show the details in the field-of-view area.

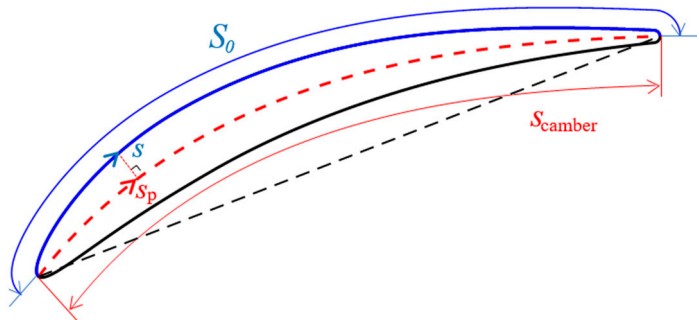

**Figure 8.** Definitions in camber projection normalization method.

The normalization process of the camber projection of the blade surface points is actually the process of solving blade thickness distribution and camber geometry. At this time, two points on the suction surface and pressure surface with the same $s_{camber}$ are symmetric along the coordinate axis of the blade thickness distribution. On the thickness distribution curve, the symmetry of the suction surface and pressure surface can be maintained from the leading edge point to the trailing edge point. Therefore, the camber projection normalization method has the best symmetry on the whole blade.

### 3.2. Effect of Thickness Distribution

The thickness distribution curve of a real blade is not strictly elliptic, and the curvature radius at the front and trailing edges is usually larger than the curvature radius at the end of the ellipse's major axis when the maximum thickness is the same. Based on the curvature radius formula of the ellipse, $r = a/\text{AR}^2$, in which $r$ is the curvature radius at the end of the major axis and $a$ is the length of the major axis, a definition of the equivalent axial ratio based on chord length and the curvature radius at leading edge point is shown as:

$$\text{ARr} = \sqrt{c/2r_{\text{LE}}} \tag{17}$$

Moreover, another definition of the equivalent axial ratio based on chord length and the maximum thickness is shown as:

$$\text{ARt} = c/H_{\max} \tag{18}$$

In the potential flow, VI near the leading edge of a straight cascade with any thickness distribution should be similar to that of an elliptic cascade with the axial ratio equal to ARr, and VI in the blade body region should be similar to that of an elliptic cascade with the axial ratio equal to ARt. The black line in Figure 9a shows a straight cascade with a stagger angle of $30°$ and a maximum thickness of 5% chord length, but with a leading edge curvature radius of 0.5% chord length. In this case, its ARr is 10 and ARt is 20. Furthermore, the standard ellipse with AR equal to 20 is shown as the blue line and the ellipse with AR equal to 10 is shown as the red line. The comparison of VI among the three geometries is shown in Figure 9b, which indicates that the elliptical cascade with ARr is the most similar to the freedom thickness straight cascade in the leading edge region, while the elliptical cascade with ARt is the most similar to the freedom thickness straight cascade in the body region. When $\beta$ changes in Figure 9c, the three leading edges keep the relative trends. Corresponding to the research content of this paper, in the actual cascade, ARr should be used as the equivalent axial ratio to calculate VI near the leading edge.

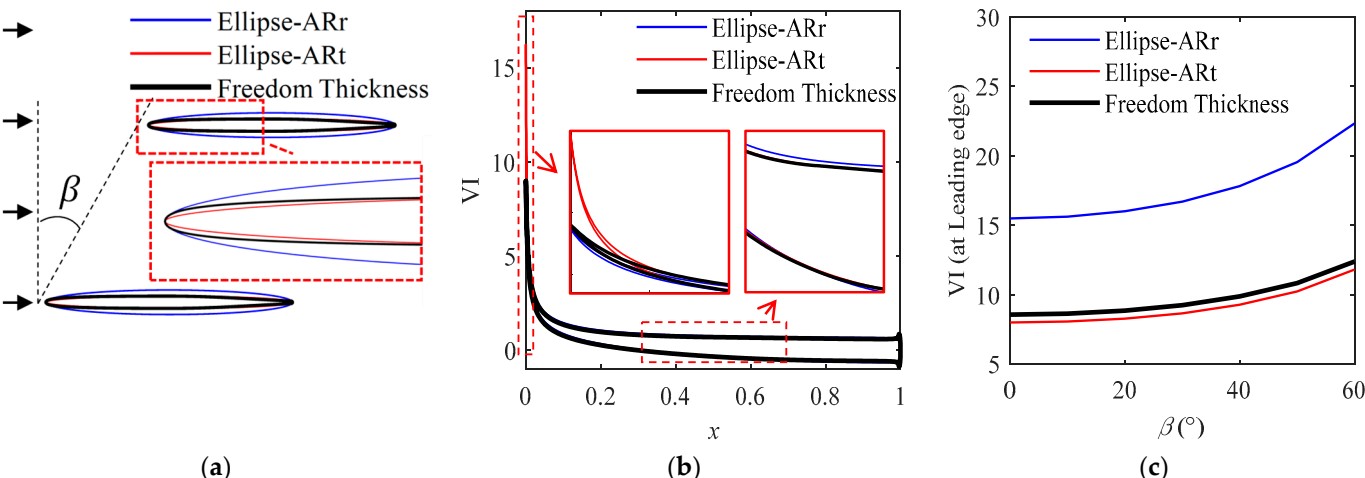

**Figure 9.** Effect of ARt and ARr for a freedom thickness straight cascade. (**a**) Geometry, (**b**) value of VI when $\beta = 30°$, (**c**) value of VI with various $\beta$ at leading edge.

### 3.3. Effect of Turning Angle of the Camber Line

The curvature of the actual camber line of the blade will affect the diffuser strength in the blade body region. If the tangent angle of the camber line at any position is defined as $\beta_x$, then according to Equations (13)–(16), VI value caused by diffuser at any position of the cascade is:

$$\text{VI}_{\text{diffusion}} = -\frac{\int_{-1}^{x} \text{VI}_{\beta,\text{corr}}(s)ds}{\int_{-1}^{1} \text{VI}_{\beta,\text{corr}}(s)ds} \cdot \frac{\sin(\alpha_1)}{\cos(\alpha_1 + (\beta_x - \beta_1)\frac{\alpha_2 - \alpha_1}{\beta_2 - \beta_1})} \tag{19}$$

However, $\text{VI}_{\text{diffusion}}$ is very small near the leading edge, meaning that the effect of the channel diffuser deceleration is not significant near the leading edge. Therefore, the influence of camber geometry on VI near the leading edge is mainly reflected in the change of the cascade stagger angle, and $\text{VI}_{\text{diffusion}}$ could be ignored when the research is focused on the leading edge region.

## 4. Numerical and Experimental Verification

The above analysis of VI starts from the potential flow solution of the plate cascade, then studies the VI distribution rule of elliptical cascade, and finally builds a VI distribution expression of an arbitrary compressor cascade by adding the influence of the thickness distribution and the camber line geometry of real blade. Due to the influence of the boundary layer, the error of the potential flow analysis results is inevitable in practice. However, the boundary layer near the blade leading edge is usually thin and has a weak influence on the mainstream flow field. Therefore, the analysis results in this paper are of high practical value. In this section, the expression of VI obtained in this paper will be verified by numerical calculations and experimental measurement.

### 4.1. Numerical Verification

The cascade bases on a typical compressor stator blade mid-span profile was used to analyze the difference between the above theoretical equation and CFD calculation near the blade leading edge, whose inlet and outlet metal angles are 51° and 6°, respectively, and the solidity is 1.44, as shown in Figure 10a. MISES was employed as the CFD tool to solve the flow field of cascade near the designed inlet flow angle (45°). In order to calculate VI, two incidence angles of ±0.5° were used, and the difference of blade surface velocity between them was used to calculate the CFD solution of VI. As the central difference scheme was adopted, the value of VI reached the second-order accuracy and met the test requirements. Figure 10b shows the difference in the values of VI between the CFD

solution and the theory estimation from Equations (11) and (16). It can be seen that $VI_\beta$ from Equation (16) can effectively predict VI in a large range of the blade surface, and there is a certain error only in the middle area of the blade suction surface. The error in the middle area of the suction surface is mainly due to the fact that the main flow velocity is higher than the incoming flow velocity, so the denominator of the relative velocity should be the local main flow velocity rather than the incoming flow velocity. $VI_{stagger,corr}$ from Equation (11) can only accurately predict the increment of angular attack velocity near the leading edge, as shown in Figure 10c. For the study of the leading edge region, $VI_{stagger,corr}$ can also be used to effectively analyze the distribution of VI and benefit from its very simple calculation process.

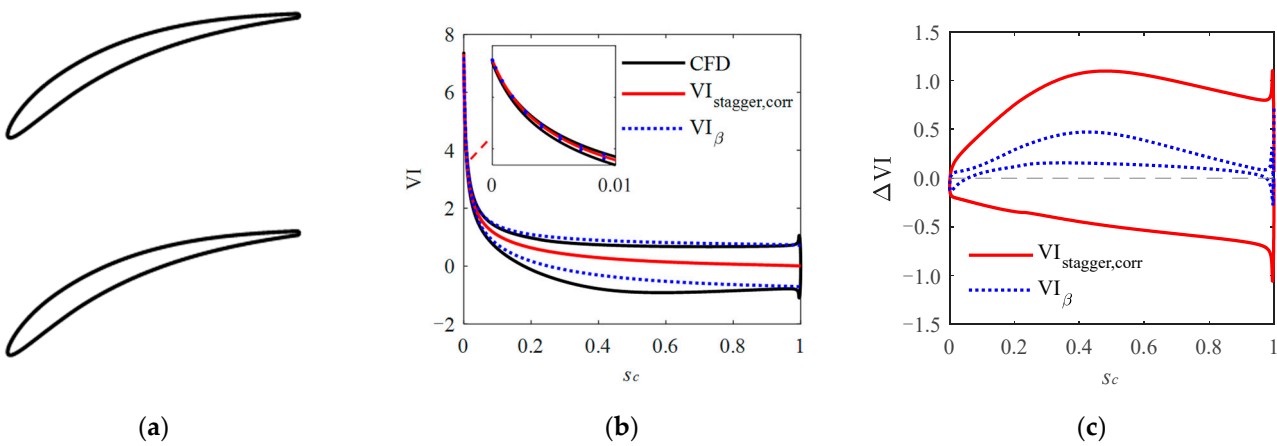

(**a**)  (**b**)  (**c**)

**Figure 10.** Numerical verification with a typical compressor stator blade cascade. (**a**) Geometry, (**b**) value of VI, (**c**) differences between calculated values and CFD result.

*4.2. Experimental Verification*

In order to further verify the accuracy of the theoretical analysis results and numerical calculation results, high spatial resolution PIV measurements have been carried out in the leading edge area to verify the changing process of blade surface velocity at different incidence angles. The experiment involved in this paper were carried out in the low-speed planar cascade facility located in Beihang University with a high-quality inlet environment (less than 1.6% turbulence) and a $420 \times 120$ mm$^2$ outlet cross-sectional area. Basic parameters of the cascade facility are shown in Table 1, in which Reynolds number is calculated based on chord length and inlet velocity.

**Table 1.** Basic parameters of the cascade facility.

| Parameters | Value |
|---|---|
| chord, c (mm) | 180 |
| solidity | 1.6 |
| inlet metal angle, $\beta_1$ (°) | 47 |
| outlet metal angle, $\beta_2$ (°) | 6 |
| designed inlet flow angle (°) | 43 |
| inlet Mach number | 0.13 |
| Reynolds number | $5.45 \times 10^5$ |
| pitch, $t$ (mm) | 112.5 |

For the PIV measurement setup, a 140 mJ/Pulse Nd:YAG laser was employed, and the light sheet thickness was approximately 1 mm. ImagerProX-4M camera with a Nikon Micro 105 mm lens and a resolution of $2048 \times 2048$ pixels (at 4 Hz recording rate) were used for image capturing, and the magnification factor of the lens was approximately 1.0. DEHS and a Laskin nozzle-based particle generator were used to generate the seeding particles. The time interval of two laser beams was 2.5 µs, and the averaged displacement

of the particles is 4.5 pixels, while the maximum displacement of the particles is 9.8 pixels. Figure 11a shows the schematic of the blade and the camera range of the PIV measurement. The physical size of the camera field-of-view shown in Figure 11 is $14.5 \times 14.5$ mm$^2$; it measured 0~8% in chord length near the leading edge. For the captured particle images shown in Figure 11b, the particle diameter is approximately 1.5–3.0 pixels, the particle distribution density is approximately 0.021–0.048 particles/pixels, and about 600 snapshots were obtained for each incidence angle condition in the experiment. While processing the pre-processed transformed images, three passes of $32 \times 32$ pixels and then three passes of $12 \times 12$ pixels interrogation windows are employed to achieve the WIDIM algorithm with a 50% overlap. As the displacement uncertainty for this algorithm is 0.05 pixels, the measured velocity uncertainty is about 1.1%. The details of the post-processing procedure was based on the method by Liu and Xu [22].

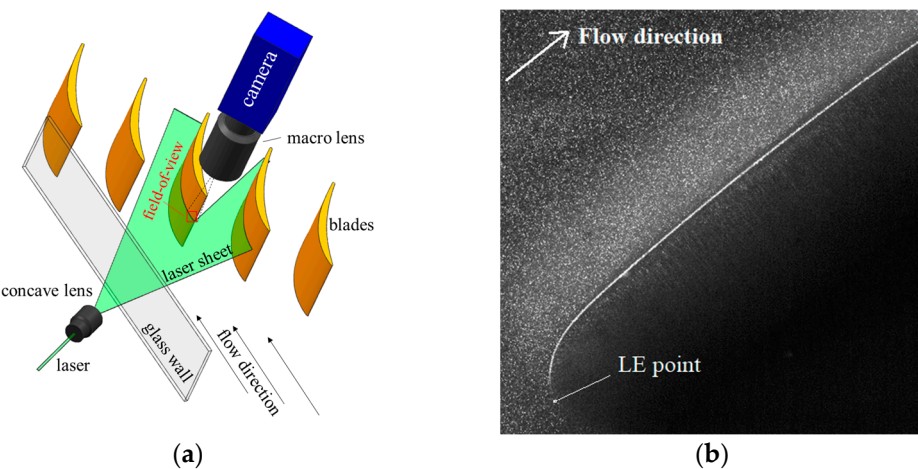

**Figure 11.** Schematic of the camera field-of-view on the cascade. (**a**) Experimental setup, (**b**) sample image.

In the experiment, velocity fields in the area of interest under six different incidence angles were obtained, and the velocity at the main flow boundary was obtained by identifying the boundary layer, as shown in Figure 12. Since the boundary layer near the leading edge at the designed incidence angle is thin, the velocity at the main flow boundary is essentially equivalent to the isentropic velocity of the blade surface. As can be seen from Figure 12, local separation bubbles appear in the leading edge area at 9.5° incidence angle, and it indicates that the boundary layer could be ignored only in the case that the incidence angle is smaller than 8.5°.

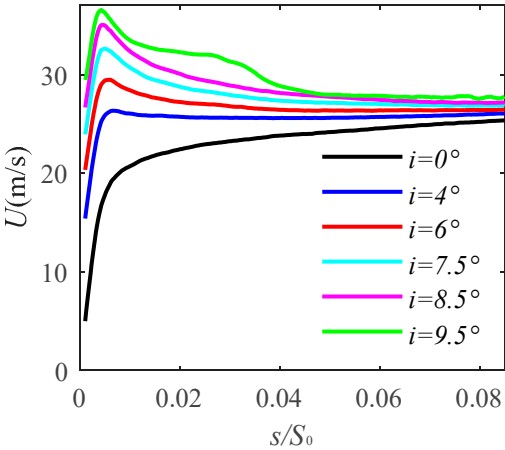

**Figure 12.** Measured velocity at the main flow boundary.

By considering the flow angle as an independent variable, the measured velocity is plotted in Figure 13a. In the figure, the higher line in the left region represents the normalized position being farther back, and all of the lines are close to the straight line. Figure 13b shows the linearity of lines in Figure 13a in different flow positions and different incidence angle ranges (black line for the small incidence angle range, while the red line for the large incidence angle range). It can be seen that in the range of the small incidence angle, the surface velocity distribution ultimately presents a linear relationship with the change of the incidence angle. Hence, when there is no serious deterioration of the boundary layer near the leading edge, the slope of the lines in Figure 13a has a physical significance, called VI, as mentioned above.

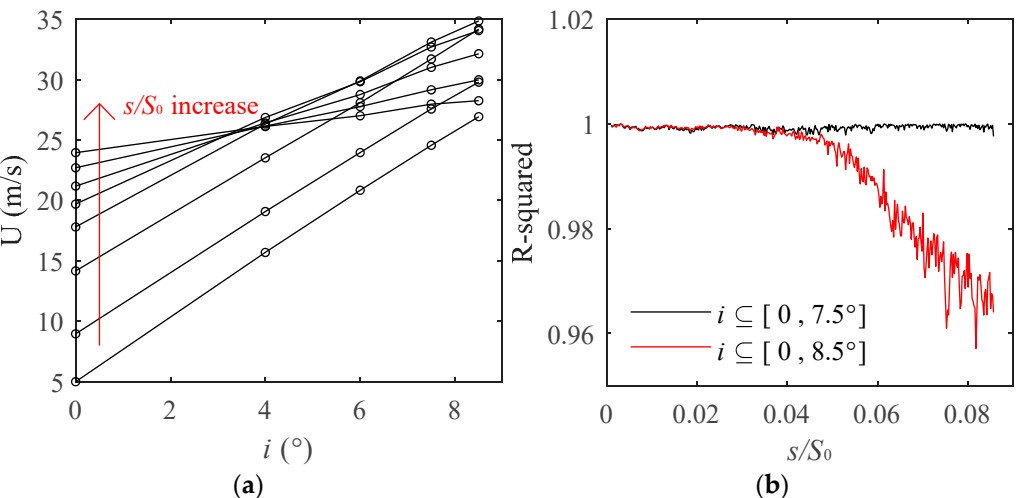

**Figure 13.** Relationship between the surface velocity and the change of incidence angle. (**a**) Velocity distribution on the incidence angle at a fixed point, (**b**) linearity of lines.

For different intervals of incidence angles, such as $[0, 4°]$, $[0, 6°]$, $[0, 7.5°]$, the value of VI measured by PIV in the leading edge region is calculated and plotted as the black dot in Figure 14. In this figure, the red line is the value calculated by CFD, and the blue line is the value calculated with Equation (16). In the experiment, VI obtained by CFD is very close to the experimental results, while VI calculated from Equation (16) is slightly different from the experimental results. However, considering that Equation (16) is an explicit analytical formula, its application is simple, so Equation (16) can be used as an effective and rapid estimation of VI near the leading edge of the blade.

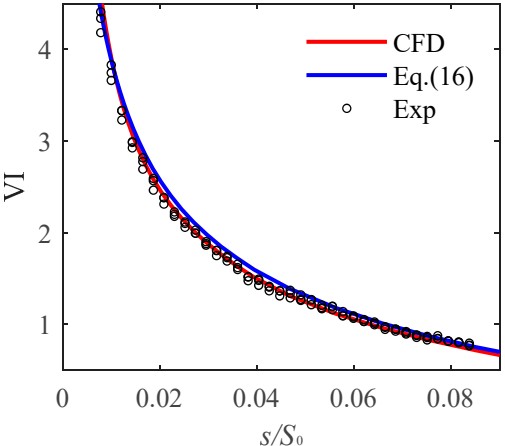

**Figure 14.** Comparison of VI Values from CFD, Equation (16), and measured by PIV.

Furthermore, both the theoretical analysis and experimental research were carried out in incompressible flow. For a subsonic compressible environment, compressibility correction can be used to further extend the calculation process in this paper [23–25].

## 5. Conclusions

In this paper, the influence of the off-designed incidence angle on the flow in the leading edge region is studied in the compressor cascade environment, and the concept of the *velocity increment on incidence angle*, VI, is raised to present the surface velocity variation with the off-designed incidence angle. With the analysis of potential theory, the explicit equation for VI is derived, and it is verified by numerical calculations and experimental measurement. The conclusions drawn are:

By employing the ratio of the value of VI between the plate cascade and the isolated plate, the infinity tendency of the current explicit solution at the leading edge point is eliminated. For the stagger angle of the cascade, the larger the stagger angle, the larger the value of VI near the leading edge region and the larger the difference between VI values of the pressure surface and suction surface in the blade body. Therefore, the equivalent pitch lengths based on $1/\cos(\beta)$ and VI caused by diffuser deceleration in the cascade passage were employed to correct the effect of the stagger angle in the explicit equation in a simplified elliptic cascade.

For the real blade profile, VI near the leading edge is similar to that of an elliptic cascade with an axial ratio equal to the equivalent axial ratio, $\sqrt{c/2r_{\mathrm{LE}}}$, and the VI value caused by camber turning can be ignored near the leading edge region. Finally, the explicit equation of VI derived in this paper depends only on the geometrical parameters of the cascade and blade.

Through numerical calculations, the equation of VI has a good coincidence accuracy near the leading edge, trailing edge, and also in the blade body region of the pressure surface. By measuring the velocity field in the leading edge area with PIV, the linear rule of the blade surface velocity on the incidence angle in the leading edge region is verified, and the solving accuracies of the equation of VI are analyzed and verified.

According to the above research, the equation of VI is suitable for a wide range of incidence angles near the leading edge of a compressor blade cascade, which greatly improve the accuracy of an explicit analytical solution of blades under all working conditions.

For feasible further research, the equation of VI could be used in a design method with lesser iterations for blade geometry by considering flow characteristics under off-designed conditions, and it could also provide a basis for research on pollution deposit positions in aeroengines by providing the calculation method for a front-stagnation position in all flow conditions. It may also be helpful in the calculation of unsteady blade force of axial compressors.

**Author Contributions:** Conceptualization, X.X. and X.Y.; methodology, X.X. and X.Y.; software, X.X.; validation, B.L. and X.Y.; formal analysis, X.X. and X.Y.; investigation, X.X., B.L., X.Y. and G.A.; resources, X.Y. and G.A.; data curation, X.X. and X.Y.; writing—original draft preparation, X.X.; writing—review and editing, G.A.; visualization, X.X.; supervision, B.L.; project administration, B.L. and X.Y.; funding acquisition, B.L. All authors have read and agreed to the published version of the manuscript.

**Funding:** This research was funded by the National Natural Science Foundation of China (Grant No. 52276025), the National Science and Technology Major Project (2017-II-0001-0013, J2019-II-0004-0024, J2019-II-0003-0023), the Advanced Jet Propulsion Innovation Center/AEAC (funding number HKCX2022-01-008), and the Fundamental Research Funds for the Central Universities.

**Data Availability Statement:** Not applicable.

**Conflicts of Interest:** The authors declare no conflict of interest. The funders had no role in the design of the study; in the collection, analyses, or interpretation of data; in the writing of the manuscript; or in the decision to publish the results.

## Abbreviations

| Symbol | Description |
| --- | --- |
| $\alpha_1$ | Inlet flow angle |
| $\alpha_2$ | Outlet flow angle |
| AR | Axial ratio of the ellipse |
| ARr | Equivalent axial ratio based on $r_{LE}$ |
| ARt | Equivalent axial ratio based on the maximum thickness |
| $\beta_1$ | Inlet metal angle |
| $\beta_2$ | Outlet metal angle |
| $\beta$ | Stagger angle |
| $c$ | Chord length |
| $H_{max}$ | Maximum thickness of the blade |
| $i$ | Incidence angle |
| $r_{LE}$ | Curvature radius at leading edge point |
| $s$ | Arc length on the blade from leading edge point |
| $s_{camber}$ | Arc length on the camber line from leading edge point |
| $s_c$ | Normalized camber projection length |
| $S_0$ | Arc length on the blade from leading edge to trailing edge |
| $T(x)$ | Ratio of flow turning angle on incidence angle |
| $t$ | Pitch length |
| $t_{eq}$ | Equivalent pitch length |
| $U$ | Velocity (scalar) |
| VI | Velocity increment on incidence angle |
| $VI_{plate,cascade}$ | VI value of the plate cascade |
| $VI_{plate,isolated}$ | VI value of the isolated plate |
| $VI_{ellipse,isolated}$ | VI value of the isolated ellipse |
| $VI_{ellipse,cascade}$ | VI value of the elliptic straight cascade |
| $VI_{stagger,corr}$ | VI value of the elliptic staggered cascade |
| $VI_{TE}$ | VI value at trailing edge point |
| $VI_{diffusion}$ | VI value caused by diffuser deceleration |
| $VI_{\beta}$ | Comprehensive VI value |

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
