# Peer review of "Velocity Increment on Incidence Angle near the Leading Edge of the Compressor Cascade"

_aerospace, doi:10.3390/aerospace10050461_

Round 1

Reviewer 1 Report

Dear Authors,

Good work!

Very well organized, and It is easier to read. I did not find any major errors in your paper, keep up the good work.

Few minor suggestions if you like to include.

1.) There is mention of error with  equation 11 and 16  in figure 10 b. From graph and your explination, it is clear with Eq. 11 we get more error vs Eq 16. Can you provide the error value or R2.

2.) From your experimental results you mention at incidence angle 9.50 you are seeing seperation bubble because of boundary layer, this makes sense. In figure 13b the R2 value increased with incidence angle 8.50. Can we assume that your proposed equation 16 has a limit of 8.50 incidence angle or it is ideal to use the equation between 0 - 8.50 incidence angle.

Author Response

Replies to the comments of Reviewer #1

Question:

  • There is mention of error with equation 11 and 16 in figure 10 b. From graph and your explication, it is clear with Eq. 11 we get more error vs Eq 16. Can you provide the error value or R2.

Reply:

  • Thanks for your reminder. Figure 10c has been added to show the error value between VIβ (or VI stagger,corr) and the CFD value. In this paper, we mainly focus on the solving accuracy near the leading-edge region.

Question:

  • From your experimental results you mention at incidence angle 9.50 you are seeing seperation bubble because of boundary layer, this makes sense. In figure 13b the R2 value increased with incidence angle 8.50. Can we assume that your proposed equation 16 has a limit of 8.50 incidence angle or it is ideal to use the equation between 0 - 8.50 incidence angle.

Reply:

  • The derivation of equation 16 is based on potential flow without considering the development of boundary layer. Therefore, we try to find out the range of incidence angle in which the boundary layer thickness in the leading-edge region can be ignored. According to the results of Figure13b, it is believed that the boundary layer thickness in the leading-edge region has no significant influence on the mainstream velocity within the range of [0, 7.5°]. For any other shape of the blade, the equation in this paper is effective in the incidence range with thin leading-edge boundary layer.

Reviewer 2 Report

The paper presents the velocity Increment on Incidence Angle near the Leading-edge of the Compressor Cascade. The paper is well written and references are adequate. I recommend this paper to publish in current form. 

Author Response

Thank you for your recognition of our research paper.

Reviewer 3 Report

The article discusses the velocity increment on incidence angle near the leading-edge of the compressor cascade. The article is well written and well structured. The presentation of the material is logical. The quality of the drawings is high. A few comments on the article.

1. The ‘Introduction’ section is not comprehensive and well-organized enough. Research background and recent development of similar studies should be reviewed, and research methods, innovation and significance of your paper should be stated. Please enrich the ‘Introduction’ section.
2. Using a list of lumped references is not very helpful for a reader (for example, ... [1-4] on page 1 or [7-10] on page 1). Assessment/justification should be provided for each reference, even it may be short.
3. Moreover, half references in the bibliography have a publication year later than 2014. More recent publications with a publication year of 2019-2023 should be added.
4. I believe that the "Notation and Abbreviations" section of the article is necessary for the convenience of readers.
5. It is necessary to add a comparison of the results obtained with the data of other authors.
6. It is necessary to indicate directions for further research on this topic in the "Conclusion" section.

In my opinion this article is interesting and worthy. However, the article has a number of shortcomings. To accept this paper for publication in the Aerospace, some improvements and revisions are required as specified.

Author Response

Question:

  • The ‘Introduction’ section is not comprehensive and well-organized enough. Research background and recent development of similar studies should be reviewed, and research methods, innovation and significance of your paper should be stated. Please enrich the ‘Introduction’ section.

Reply:

  • Thanks for your professional comments. The introduction was rewritten carefully to ensure a clear logic.

Question:

  • Using a list of lumped references is not very helpful for a reader (for example, ... [1-4] on page 1 or [7-10] on page 1). Assessment/justification should be provided for each reference, even it may be short.

Reply:

  • Thanks for your professional comments. The introduction was rewritten carefully, and the contributions of each reference have been written in detail.

Question:

  • Moreover, half references in the bibliography have a publication year later than 2014. More recent publications with a publication year of 2019-2023 should be added.

Reply:

  • After re-searching the references and research status, we added the recent studies on the leading-edge design methods. However, the study about potential theories for cascade flow have stagnant for decades, no more recently references could be considered. We hope that our research on the explicit potential solution of the cascade can provide new ideas for blade design.

Question:

  • I believe that the "Notation and Abbreviations" section of the article is necessary for the convenience of readers.

Reply:

  • As suggested, “Notation and Abbreviations” section has been added, and been placed after the “Conclusions”.

Question:

  • It is necessary to add a comparison of the results obtained with the data of other authors.

Reply:

  • Thanks for your professional comments. We tried to find data from other researchers, but did not find the experimental data which contains the detailed measured velocity near the leading-edge and also the detailed geometric parameters. We would keep searching the references to improve our paper.

Question:

  • It is necessary to indicate directions for further research on this topic in the "Conclusion" section.

Reply:

We believe that the research of this paper has wide application prospect. In the feasible further research, the equation of VI could be used in iteration-less blade geometry design by considering flow characteristics under off-design conditions, and it could also provide a basis for the research about pollution deposit position in aeroengine by giving the calculation method of front stagnation position in any flow condition. It may also be helpful in the calculation of unsteady blade force of axial-compressor.

Reviewer 4 Report

Overall, the paper is not ready for publication and needs major rewrite.

Please see my attached suggestions to improve the manuscript.

Author Response

Question:

  • Literature survey is dated and old. Please include more of the recently published work.

Reply:

  • Thanks for your professional comments. After re-searching the references and research status, we added the recent studies on the leading-edge design methods. However, the study about potential theories for cascade flow have stagnant for decades, no more recently references could be considered. We hope that our research can provide new ideas for blade design.

Question:

  • Figure 14- who did the experiment? If you did, where are the details? Line 305 indicates measurements will be carried out- it is confusing- have you taken the measurements or it will be? What is the uncertainty? Show a schematic of the whole setup in figure 11 to illustrate how the flow and measurements were carried out.

Reply:

  • Thanks for your professional comments. The details of the experiment setup have been added in Section 4.2. The measurement uncertainty has been analyzed in our previous article ( Ref [22] in the revision). With the displacement uncertainty for the PIV algorithm is 0.05 pixels, the measured velocity uncertainty is about 1.1%, in this paper. Figure 11 has been replotted to contain the whole setup.

Question:

  • Identify the airfoil shape in the diagram. Put the spacing and orientation numbers in the figure itself so that reader does not need to hunt for information, Fig 5, Fig 7b. Figure 8 needs to be combined others and numbers provided. Figure 11 is too simple to have its own- combine with others.

Reply:

  • As suggested, the cascade shapes and other information have been added onto the diagrams to make them easier to read (involve Figure 5a, 7, 8, 9c, 10c ).

Question:

  • Figure 9 is not done right. Improve it with addition of curved airfoil and different angles.

Reply:

  • As suggested, the effect of different stagger angles on the value of VI at leading-edge have been plotted as Figure 9c. For the curved airfoil, we think Figure 10 have the appropriate expression.

Question:

  • There are several equations on VI calculation and confusing for the reader. Identify the equation used for the figure in figure caption for the ease of reading.

Reply:

  • Thanks for your professional comments. As the equations are always wordy, we perfected the wording of the captions.

Round 2

Reviewer 3 Report

All comments have been corrected. The article may be accepted for publication.

Reviewer 4 Report

Authors have addressed my concerns and added experimental figures to clarify the content. The paper is now acceptable for publication. They opted not to cite the samples provided in prior review comments. It could have illustrated this work's differences from the recent trend.

A suggested improvement is to add reference numbers to all the cited works, especially when they are referenced in figures and in separate paragraphs. For example, Baddoo's [19] work has not been referenced with number in Figure 6 or in the text body at several places.